# Is the Tyme Wear Smart Shirt Reliable and Valid at Detecting Personalized Ventilatory Thresholds in Recreationally Active Individuals?

**DOI:** 10.3390/ijerph19031147

**Published:** 2022-01-20

**Authors:** Aaron H. Gouw, Gary P. Van Guilder, Gillian G. Cullen, Lance C. Dalleck

**Affiliations:** Recreation, Exercise & Sport Science, Western Colorado University, Gunnison, CO 81231, USA; aaron.gouw@western.edu (A.H.G.); gvanguilder@western.edu (G.P.V.G.); gillian.cullen@western.edu (G.G.C.)

**Keywords:** aerobic threshold, anaerobic threshold, exercise prescription, threshold detection, threshold-based training

## Abstract

The aim of this study was to determine the extent to which the Tyme Wear smart shirt is as reliable and valid in detecting personalized ventilatory thresholds when compared to the Parvo Medics TrueOne 2400. In this validation study, 19 subjects were recruited to conduct two graded exercise test (GXT) trials. Each GXT trial was separated by 7 to 10 days of rest. During the GXT, gas exchange and heart rate data were collected by the TrueOne 2400 (TRUE) in addition to the ventilation data collected by the Tyme Wear smart shirt (S-PRED). Gas exchange data from TRUE were used to detect ventilatory threshold 1 (VT1) and ventilatory threshold 2 (VT2). TRUE and S-PRED VT1 and VT2 were compared to determine the reliability and validity of the smart shirt. Of the 19 subjects, data from 15 subjects were used during analysis. S-PRED exhibited excellent (intraclass correlation coefficient—CC > 0.90) reliability for detection of VT1 and VT2 utilizing time point and workload and moderate (0.90 > ICC > 0.75) reliability utilizing heart rate. TRUE exhibited excellent reliability for detection of VT1 and VT2 utilizing time point, workload, and heart rate. When compared to TRUE, S-PRED appears to underestimate the VT1 workload (*p* > 0.05) across both trials and heart rate (*p* < 0.05) for trial 1. However, S-PRED appears to underestimate VT2 workload (*p* < 0.05) and heart rate (*p* < 0.05) across both trials. The result from this study suggests that the Tyme Wear smart shirt is less valid but is comparable in reliability when compared to the gold standard. Moreover, despite the underestimation of S-PRED VT1 and VT2, the S-PRED-detected personalized ventilatory thresholds provide an adequate training workload for most individuals. In conclusion, the Tyme Wear smart shirt provides easily accessible testing to establish threshold-guided training zones but does not devalue the long-standing laboratory equivalent.

## 1. Introduction

The development and use of consumer wearable technology has grown a considerable amount in the past five years. In 2016, the global market for wearables was valued at USD 6.18 billion. In 2019, the market was valued at USD 32.63 billion [1]. The global market size for wearable technology is expected to grow by 15.9% from 2020 to 2027 [1]. Typically, the goal of wearable technology has been to easily provide previously unknown biometric information regarding the user’s physiology or lifestyle behavior. This information can be used to track fitness, guide training, or provide general health data such as heart rate, heart rate variability, oxygen saturation, and many other metrics. While the majority of wearable technology is typically used by consumers, there are wearable technologies that are used in the medical field with clinical populations. Even as the adoption of wearable technology continues to grow, there are continuing challenges associated with cost, poor data representation, lack of usability, and excessive information [2]. However, the greatest challenges that appear to plague wearable technologies are the reliability and validity of the biometric data and thus the accuracy of the analysis and conclusions reported by these wearable devices.

In the context of wearable technology, reliability is defined as the consistency of the data collected from a measurement, which is also known as test–retest reliability [3]. To measure and test the reliability of the wearable, multiple measurements must be taken and compared to evaluate the ability of the wearable to consistently produce measurements. Validity is defined as the extent to which the data collected from a measurement accurately represent the reality of the variable(s) intended to be measured. In order to test validity, the wearable technology must be compared to the gold standard. For example, a smart watch that measures heart rate would have the validity of the heart rate measurement compared to an ambulatory electrocardiogram in order to determine the validity of the wearable. In addition, to support the analysis of reliability and validity of wearable technology, previous studies have utilized key statistical tests. The statistical analyses include: (1) Bland–Altman analysis and plots; (2) intraclass correlation coefficient; (3) typical error [4].

While there are many physiological measures that can be measured to determine and support training related to exercise performance, ventilatory thresholds have been thoroughly utilized in both performance and clinical populations and supported with extensive evidence of their effectiveness to guide training and measure performance [5,6,7]. Determination of ventilatory thresholds is a function of minute ventilation (VE) and volume of oxygen consumed (VO_2_) or VE and volume of carbon dioxide produced (VCO_2_). The ventilatory thresholds, ventilatory threshold 1 (VT1) and 2 (VT2), vary depending on the individual, correspond to specific exercise intensities or workload, and demarcate shifts in exercise metabolism. Using these measurements, there are three primary ways that the ventilatory thresholds can be calculated: (1) ventilation curve method; (2) V-slope method; (3) ventilatory equivalent method [8].

With the continued advancement in hardware and software development, devices can now measure and collect physiological data, which was once only possible in the laboratory environment. An example of this unique measurement in wearable technology is VE. The collection of VE data can be used to determine ventilatory threshold in its users [9]. Tyme Wear has developed a smart shirt and app that purportedly measures VE to determine ventilatory thresholds. The data can then be used to provide training recommendations based upon the first and second ventilatory thresholds as well as performance prediction for running races such as marathons. As with all wearable technologies, the Tyme Wear device should be tested for reliability and validity through comparison to the gold standard in order to be considered a viable product for the consumer.

The purpose of this study was to determine the reliability and validity of the Tyme Wear smart shirt compared to the Parvo Medics TrueOne 2400 in determining ventilatory thresholds in recreationally active men and women. As such, the question to be addressed is as follows: To what extent is the Tyme Wear smart shirt reliable and valid in determining ventilatory thresholds when compared to the gold standard?

## 2. Materials and Methods

Potential subjects were recruited via email or word of mouth. Subjects were Western Colorado University (WCU) students and staff as well as members of the Gunnison, Colorado community. Inclusionary criteria included recreationally active persons, male or female, between the ages of 18 and 65 years of age. Exclusion criteria included pulmonary and cardiovascular conditions as well as any physical conditions that may be agitated or exacerbated due to a running, maximal graded exercise test (GXT).

### 2.1. Experimental Design

The experimental flow chart outlining the study design is shown in Figure 1. This validation study was conducted on the WCU campus in the High Altitude Performance Lab (HAP Lab) in Gunnison, Colorado, United States (2348 m). The study consisted of two trials of a GXT that were separated by 7 to 10 days to allow for adequate rest between sessions. Exact rest days were determined based on scheduling availability of the researcher and subject. Two days prior to the initial GXT in trial 1, subjects were provided pre-test instruction and allowed to keep a copy of the instructions as a reminder for the second test. Pre-test questionnaires (Appendix A) were completed prior to all tests to screen subjects for readiness. Subjects whose GXT did not meet the criteria of completion were given the option to: (1) complete an additional GXT 7–10 days after the test; (2) choose to no longer participate in the study.

### 2.2. Procedures

#### 2.2.1. Pre-Test Preparation

Subjects were given a pre-test questionnaire that determined the subject’s GXT readiness. The questionnaire was developed based upon variables that could influence the outcome of the GXT. Furthermore, this questionnaire allowed the subjects to replicate preparatory conditions (exercise, diet, etc.) for the second GXT session. If subject met all requirements for testing, they were prepared for the GXT. In order to collect heart rate data, subjects were fitted with a chest strap heart rate monitor (WearLink, Polar USA, Worcester, MA, USA). Subjects were then provided a smart shirt (Tyme Wear smart shirt, Tyme Wear, Boston, MA, USA) that was fitted according to the sizing guides provided by Tyme Wear. The Tyme Wear pod was subsequently attached to the hub of the smart shirt (refer to Figure 2 for external layout of shirt).

#### 2.2.2. Anthropometric Data Collection

Prior to the GXT, the subject’s height (cm) and weight (kg) were measured using a digital stadiometer (WB-3000, Tanita Corporation of America, Arlington, IL, USA). Age was also recorded during this time.

#### 2.2.3. Graded Exercise Test (GXT)

The protocol for the GXT was developed by Tyme Wear. Directions for the GXT were administered via a smart phone app and used to guide increases in exercise intensity during the GXT. Once the subject was connected to the metabolic cart (TrueOne 2400, Parvo Medics, Salt Lake City, UT, USA) and connection between the app and smart shirt was confirmed, the GXT was initiated. All GXTs occurred on a motorized treadmill (Fitnex Fitness Equipment, Dallas, TX, USA). Heart rate was continuously measured and collected via the TrueOne2400 with a Polar heart rate sensor antenna (GymLink, Polar USA, Worcester, MA). Additional heart rate data were measured and collected separately on two polar heart rate monitor watches (Polar FS1, Polar USA, Worcester, MA, USA) at 30 s intervals in the event the TrueOne 2400 failed to collect heart rate data or to substitute anomalies in the heart rate data collected by the TrueOne2400. Prior to the GXT, breathing data from three full breaths (three full expirations and three full inspirations) were collected by the smart shirt and TrueOne2400. The GXT protocol began with one minute of standing rest while ventilatory data were collected by the smart shirt and gas exchange data collected by the TrueOne 2400. Subsequently, the subject was required to walk, as a warmup, at 3.0 mph for three minutes. Following this warmup, the subject was required to begin running at 3.3 mph, and running speed was increased 0.3 mph every 30 s until volitional fatigue. Then, the subject was asked to stand upright for one additional minute after cessation of the GXT while the smart shirt and metabolic cart continued to collect ventilation and gas exchange data, respectively.

#### 2.2.4. Criteria for VO_2_max Test

The criteria for a true VO_2_max test were established for this study according to previous research [10]. The following criteria were used to determine whether a subject reached VO_2_max: (1) a respiratory exchange ratio (RER) of 1.10 or greater for men or women aged 20–49 and 1.05 or greater for men or women aged 50–64; (2) plateau of oxygen consumption less than or equal to 150 mL O_2•_min^−1^; (3) a rating of perceived exertion greater than eight (on a scale of 1–10; converted from Borg scale).

#### 2.2.5. Identification of Ventilatory Thresholds Using Gas Exchange Data

The gold standard VT1 and VT2 were identified from gas exchange data through use of visual inspection. Visual assessment was completed by two trained exercise physiologists. In case of disagreement, assessments were reevaluated until agreement was reached. This method for assessment and determination is consistent with past work [11,12]. Data averaged at 15 s intervals were plotted according to the appropriate variables for VT1 (VE/VO_2_) and VT2 (VE/VCO_2_). VT1 was identified based on the criteria of an increase in VE/VO_2_ with VE increasing non-linearly in addition to the nonconcurrent increase in VE/VCO_2_. VT2 was identified based on the concurrent increase in VE/VO_2_ and VE/VCO_2_.

### 2.3. Statistical Analyses

Parameters (timepoint detection, heart rate, and workload) at VT1 and VT2 for the smart shirt (S-PRED) and TrueOne 2400 (TRUE) were compared using the paired *t*-test to test for significant differences. A *p*-value less than 0.05 rejects the null hypothesis and suggests a statistically significant difference between groups. A *p*-value greater than 0.05 accepts the null hypothesis suggests a nonsignificant difference between groups. Analyses were conducted using Microsoft Excel software (Microsoft Excel, Microsoft Corporation, Redmond, WA, USA).

To determine the reliability of the Tyme Wear smart shirt, predicted VT1 and VT2 values from the two trials of each subject were compared to the equivalent data derived from the TrueOne 2400. To establish the reliability of the Tyme Wear smart shirt, the following statistical analyses were used: coefficient of variation (CV), typical error (TE), and intraclass correlation coefficient (ICC). Ranges for ICC are interpreted as excellent (ICC > 0.9), moderate (0.90 > ICC > 0.75), fair (0.75 > ICC > 0.50), and poor (0.5 > ICC) [13]. Reliability analysis was conducted using Microsoft Excel software (Microsoft Excel, Microsoft Corporation, Redmond, WA, USA).

To determine the validity of VT1 and VT2 determined by the Tyme Wear smart shirt, VT1 and VT2 determined by gas exchange data from the TrueOne 2400 were compared to the predicted VT1 and VT2 values determined by the ventilation data from the smart shirt. To compare the validity of the Tyme Wear smart shirt to the TrueOne2400, Bland–Altman 95% limits of agreement were used to quantify the agreement (bias ± random error (1.96 × SD)) [14]. Analyses and plots were conducted and created with GraphPad Prism software (Prism, GraphPad Software, San Diego, CA, USA).

## 3. Results

### 3.1. Subjects

Nineteen recreational active men and women volunteered for this study. Of the 19 subjects tested, four subjects’ data were excluded due to equipment error or data loss during testing. All subjects met the criteria for a valid VO_2_max test. Relevant characteristics are described in Table 1.

### 3.2. Reliability

Measures of reliability are described in Table 2 and Table 3 for TRUE and S-PRED, respectively. Reliability measures include CV, TE, and ICC. Furthermore, the measures for reliability at VT1 and VT2 are presented across the three parameters: time point, heart rate, and workload. Table 2 shows that TRUE has excellent reliability in detecting VT1 and VT2 utilizing time point, heart rate, and workload. Similarly, Table 3 shows that S-PRED has excellent reliability in detecting VT1 and VT2 utilizing time point and workload. However, Table 3 shows that S-PRED has moderate reliability in detecting VT1 and VT2 utilizing heart rate.

### 3.3. Validity

Averages for time point, workload, and heart rate values at VT1 and VT2 during GXT trials 1 and 2 for TRUE and S-PRED are presented in Table 4. Paired *t*-tests revealed significant mean differences for heart rate at VT1 (*p* = 0.02) and VT2 (*p* = 0.02) between TRUE and S-PRED during trial 1. Additionally, there were significant mean differences between TRUE and S-PRED for workload (*p* = 0.02) and time point detection (*p* = 0.02) at VT2 during trial 1. During trial 2, there were significant mean differences between TRUE and S-PRED time point detection (*p* = 0.01), workload (*p* = 0.01), and heart rate (*p* = 0.02) for VT2.

Furthermore, Table 5 presents the maximum, average, and minimum heart rate as a percentage of achieved peak heart rate at VT1 and VT2 estimated by TRUE and S-PRED for the cohort. The paired *t*-test suggests significant mean differences between TRUE and S-PRED in the average % of max heart rate for VT1 in trial 1 (*p* = 0.02) as well as VT2 in trial 1 (*p* = 0.01) and trial 2 (*p* = 0.02). There was no significant difference between TRUE and S-PRED in the average % of max heart rate at VT1 for trial 2 (*p* > 0.05).

Similarly, Table 6 presents the maximum, average, and minimum workload as a percentage of maximum workload at VT1 and VT2 estimated from TRUE and S-PRED for the cohort. In Table 6, the paired *t*-test suggests significant mean differences between TRUE and S-PRED in the average % of max workload speed for VT2 in trial 1 (*p* = 0.01) and trial 2 (*p* = 0.01). There were no significant differences between TRUE and S-PRED in the average % of max workload at VT1 for trial 1 and trial 2 (*p* > 0.05).

### 3.4. Bland–Altman Plots and Analyses

Bland–Altman plots indicating the mean differences in GXT VT1 and VT2 time point detection between TRUE and S-PRED and levels of agreement with 95% confidence intervals (CIs) are illustrated in Figure 3 and Figure 4. For trial 1 VT1 time point detection, the mean difference between TRUE and S-PRED was 0.3 ± 0.9 min (95% CI, −1.4 to 2.0 min). For trial 2 VT1 time point detection, the mean difference between TRUE and S-PRED was 0.4 ± 1.4 min (95% CI, −2.2 to 3.1 min). For trial 1 VT2 time point detection, the mean difference between TRUE and S-PRED was 0.82 ± 1.46 min (95% CI, −2.0 to 3.7 min). For trial 2 VT2 time point detection, the mean difference between TRUE and S-PRED was 1.2 ± 1.8 min (95% CI, −2.4 to 4.8 min).

Bland–Altman plots indicating the mean differences in GXT VT1 and VT2 workload between TRUE and S-PRED and levels of agreement with 95% CIs are illustrated in Figure 5 and Figure 6. For trial 1 VT1 workload, the mean difference between TRUE and S-PRED was 0.2 ± 0.5 mph (95% CI, −0.8 to 1.2 mph). For trial 2 VT1 workload, the mean difference between TRUE and S-PRED was 0.3 ± 0.8 mph (95% CI, −1.3 to 1.8 mph). For trial 1 VT2 workload, the mean difference between TRUE and S-PRED was 0.5 ± 0.9 bpm (95% CI, −1.2 to 2.2 mph). For trial 2 VT2 workload, the mean difference between TRUE and S-PRED was 0.8 ± 1.1 mph (95% CI, −1.5 to 3.0 mph).

Bland–Altman plots indicating the mean differences in GXT VT1 and VT2 heart rate between TRUE and S-PRED and levels of agreement with 95% CIs are illustrated in Figure 7 and Figure 8. For trial 1 VT1 heart rate, the mean difference between TRUE and S-PRED was 4.0 ± 7.2 bpm (95% CI, −10.1 to 18.1 bpm). For trial 2 VT1 heart rate, the mean difference between TRUE and S-PRED was 4.3 ± 11.1 bpm (95% CI, −17.6 to 26.1 bpm). For trial 1 VT2 heart rate, the mean difference between TRUE and S-PRED was 5.3 ± 8.6 bpm (95% CI, −11.7 to 22.2 bpm). For trial 2 VT2 heart rate, the mean difference between TRUE and S-PRED was 8.2 ± 13.7 bpm (95% CI, −18.7 to 35.1 bpm).

## 4. Discussion

### 4.1. Primary Findings

The aim of this study was to evaluate the reliability and validity of the Tyme Wear smart shirt to predict VT1 and VT2 when compared to the gold standard determined by open-circuit indirect calorimetry. Based on the findings from this study, the ventilatory threshold data suggest that the Tyme Wear smart shirt (S-PRED) has similar reliability to the gold standard (TRUE). However, the ventilatory threshold data suggest that S-PRED is less valid relative to TRUE. Consequently, there is a tendency for S-PRED to underestimate the workload and heart rate values corresponding to VT1 and VT2 compared to TRUE VT1 and VT2. These novel and encouraging findings provide the necessary data to establish the foundation to determine the efficacy of wearable technologies to guide threshold-based training.

### 4.2. Reliability: Cohort Level

Data from the current study support previous findings of TRUE’s *excellent* reliability with ICC values greater than 0.9 across all trials for timepoint, heart rate, and workload corresponding to VT1 and VT2 [15,16,17]. Similarly, S-PRED also produced *excellent* reliability utilizing time point to determine VT1 and VT2; *moderate* reliability in determining VT1 and VT2 heart rate; and *excellent* reliability when using workload to determine VT1 and VT2.

Another important finding from this study is the expected technical error values, a combination of both biological error and measurement error, for VT1 and VT2. For example, TRUE data suggest the technical error, as CV, for VT1 and VT2 workload to be 6.8% and 4.9%, respectively. Furthermore, TRUE data suggest the technical error, as CV, for heart rate at VT1 and VT2 to be 3.2% and 2.4%, respectively. The results in the current study for CV are within similar ranges to those found elsewhere where CV for VT1 workload and heart rate were 5.2% and 4.3%, respectively, while CV for VT2 workload and heart rate were 2.4% and 4.2%, respectively [18]. The finding from the current study for CV VT1 workload is also similar to that suggested by Dickhurth and colleagues (CV = 5.8%) [19]. Interestingly, in a study by Gasparini Neto and colleagues, the suggested heart rate CV values for non-athletes at VT1 and VT2 were 11.8% and 6.6%, respectively, while elite athletes’ heart rate CV values for VT1 and VT2 were 3.5% and 5.1%, respectively [20]. The authors suggest that the higher CV in the non-athletes could have been due to large inter-stage increases in intensity during the GXT which may not be appropriate for this population. As such, it appears the differences in published CV values can be attributed to the GXT protocol, the population under study, equipment used to measure gas exchange, and methods used to detect VT1 and VT2 [21].

### 4.3. Validity: Cohort Level

The data presented in Table 4 suggest that S-PRED underestimates the timepoint, workload, and heart rate values corresponding to VT1 and VT2 when compared to TRUE. Furthermore, the bias and wide limits of agreement revealed in the Bland–Altman plots suggest that the agreement between TRUE and S-PRED is less than optimal for VT1 and even less so for VT2. The difference in workload is approximately 2–3% and 5–9% between TRUE and S-PRED VT1 and VT2, respectively, as observed in Table 6. For heart rate, the difference is approximately 2% and 2–5% between TRUE and S-PRED VT1 and VT2, respectively, as observed in Table 5. This translates to a workload difference of 0.2–0.3 and 0.5–0.9 mph between TRUE and S-PRED VT1 and VT2, respectively. For heart rate, this translates to a difference of 3.7 and 3.7–9.2 bpm between TRUE and S-PRED VT1 and VT2, respectively. As such, it appears that for the cohort in our study, for every 0.3 mph increase, heart rate increases concomitantly by 5–10 bpm.

However, is the underestimation of S-PRED heart rate and workload outside the physiological range for VT1 and VT2? A study by Gasparini Neto et al. [20] suggested that for non-athletes, VT1 occurred at 76.0 ± 5.4% of heart rate max while elite athletes’ VT1 occurred at 85.5 ± 4.2%. Their data also suggests that VT2 for non-athletes occurred at 91.4 ± 4.0% of heart rate max, while VT2 for elite athletes occurred at 94.9 ± 2.4% of heart rate max [20]. In persons with chronic stroke, Boyne et al. [6] suggested that VT1 occurred at 80% of peak heart rate, while the ranges for effectively exercising at VT1 ranged from 55% to 80% of peak heart rate. In another study, Cerezuela-Espejo and colleagues suggested that the 95% confidence interval for VT1 and VT2 heart rate were 77–81% and 91–93% of max heart rate, respectively, and that VT1 and VT2 workload were 59–65% and 84–87% of maximum achieved speed, respectively [22]. Taken together, the ranges for VT1 and VT2 suggested in the aforementioned studies suggest that the underestimation observed in Table 5 and Table 6 for S-PRED VT1 may not be physiologically significant, but the underestimation for S-PRED VT2 may be physiologically significant. As such, it can be postulated that for our cohort, VT1 heart rate can range within 15–20 bpm and remain within the exercise intensity range of VT1, while VT2 heart rate range can be within 12 bpm and remain within the exercise intensity range of VT2.

### 4.4. S-PRED Validity: Individual Level

While accounting for the individual variations that fall beyond the aforementioned ranges for VT1 and VT2 workload and heart rate, which can be observed in the Bland–Altman plots, S-PRED may be effective in identifying workloads corresponding to VT1 and VT2 for approximately 83% and 57% of the cohort and similar populations, respectively. Additionally, S-PRED may be effective at detecting heart rate values related to VT1 and VT2 for approximately 86% and 67% of this cohort, respectively. However, the underestimation appears to be exacerbated for a select number of subjects in this cohort. Specifically, these underestimations for S-PRED heart rate were exacerbated in one and three individuals for VT1 and VT2, respectively, in trial 1. The ranges of the severe underestimations in trial 1 were 8.4% and 7.1–12.3% below TRUE for VT1 and VT2, respectively. Furthermore, these underestimations for S-PRED heart rate were exacerbated in three and seven individuals for VT1 and VT2, respectively, in trial 2. The ranges of these more severe underestimations in trial 2 were 8.8–15.0% and 7.21–14.7% below TRUE for VT1 and VT2, respectively. In terms of workload, the underestimations were exacerbated in one and five individuals for VT1 and VT2, respectively, for trial 1. The ranges of these more severe underestimations in trial 1 were 11.6% and 11.6–22.0% below TRUE for VT1 and VT2, respectively. Additionally, in trial 2, the underestimations were exacerbated in four and eight individuals for the workloads corresponding to VT1 and VT2, respectively. The ranges of these more severe underestimations in trial 2 were 12.5–16.1% and 10.3–27.3% below TRUE for VT1 and VT2, respectively.

Upon closer inspection of the Bland–Altman plots, there are four particular subjects whose underestimation is consistently larger than the others for workload and heart rate at VT1 and VT2 across both trials 1 and 2. Specifically, in Figure 6 and Figure 7, it appears that these four subjects that had the greatest difference between S-PRED and TRUE and also had higher fitness levels (VO_2_max > 60 mL⋅kg^−1^⋅min^−1^ for males and > 50 mL⋅kg^−1^⋅min^−1^ for females) when compared to the rest of the cohort, whereby VT1 and VT2 workload occurred at higher intensities. One possible explanation for this is that the algorithms used to model VT1 and VT2 are not well suited for those users with greater fitness levels and may be unable to accurately detect VT1 and VT2 workloads for these types of individual.

### 4.5. Potential Explanation for Reduced Validity of the Tyme Wear Smart Shirt

There are several reasons why S-PRED is less valid when compared to the TRUE. There are two points where error can arise: (1) data collected by the sensors and (2) the accuracy of the algorithms in correctly detecting ventilatory thresholds. Regarding the sensor data collection, S-PRED utilizes proxy measurements (sensor stretch and rate of stretch) to replicate the equivalent data (tidal volume and respiratory rate, respectively). Furthermore, we are unaware of the inter-unit measurement error of the sensors. Additionally, we are unfamiliar with the algorithms used to determine VT1 and VT2. Without the ability to compare S-PRED VE and TRUE VE in liters per minute, it is difficult to conduct an equivalent comparison and to determine whether the S-PRED underestimations of VT1 and VT2 are due to the sensors or algorithms. If the underestimation of VE is the primary cause of the underestimation of VT1 and VT2 for S-PRED, it is likely that the sensor is underestimating tidal volume. Based on the understanding of the technology, respiratory rate would be easily determined, but the estimations for tidal volume may be more difficult as the fit of the shirt and the sensor’s sensitivity to the expansion of the thoracic cavity during a respiratory cycle creates potential for measurement error. However, it is plausible that sensor and algorithm error are contributing to the underestimation.

Finally, the cost of the TrueOne2400 is magnitudes greater in cost than the Tyme Wear smart shirt. Simply put, the smart shirt is not expected to be as accurate or reliable as the TrueOne2400. However, this does not insinuate that the Tyme Wear smart shirt is ineffective at estimating VT1 and VT2. Rather, it appears that the sensors, algorithms, or both may need further refinement to improve capture accuracy and ventilation data processing to establish greater reliability and validity when detecting ventilatory thresholds for a segment of the population with high levels of fitness.

### 4.6. Value and Efficacy of the Tyme Wear Smart Shirt to Detect Ventilatory Thresholds

In this current study, the primary concern with the underestimation of the S-PRED is that the suggested training intensity for VT1 and VT2 may not be sufficient to elicit the necessary and desired adaptive response to prescribed exercise and workouts. Even though the shirt may underestimate the exercise intensities, it can be argued that personalized, structured training within specified, albeit underestimated for S-PRED, workload zones is superior to standardized training [5,11,12,23]. Thus, the question arises: Is the underestimation by S-PRED significant enough to detrimentally impact training and performance? While a future study should address this question, it is possible to speculate on the outcomes. The consequence of training below the true physiological VT1 and VT2 means that users may not receive the adequate dose of exercise stress needed to elicit improvements in exercise performance. However, depending on the individual utilizing the smart shirt, an untrained individual may observe improvements in their exercise performance simply due to the introduction of exercise stress. Contrastingly, an elite athlete, for example, may need more precision than the smart shirt offers, specifically for VT2, as sufficient exercise stress at specific intensities is required to improve performance in a well-trained system.

### 4.7. Strength of Study and Additional Findings

The primary strength of the present study is that it is the first investigation, to our knowledge, of the reliability and validity of a wearable technology that measures respiratory rate and tidal volume to calculate VE in order to detect VT1 and VT2. Our novel findings provide the necessary preliminary data to establish the potential for the Tyme Wear smart shirt to improve performance by optimizing training with the use of personalized VT1 and VT2. Traditionally, ventilatory thresholds are determined through gas exchange data as functions of VO_2_, VCO_2_, and VE, whereby plots of VE/VO_2_ are used to determine VT1 and plots of VE/VCO_2_ are used to determine VT2 [24]. Examples of these methods include the ventilation curve method, V-slope method, and ventilatory equivalents method [8]. Furthermore, there are field-based methods such as the talk test which can determine VT1 with high reliability and validity [25,26,27]. However, our lab recently conducted a retrospective analysis and found that the sole use of VE was a sufficient and viable way to estimate VT1 and VT2 [9]. This finding supports the paradigm that the Tyme Wear smart shirt has the potential to accurately detect VT1 and VT2 by exclusively using VE.

Additionally, our study provides evidence to support the ranges for percent of maximum heart rate and workload at VT1 and VT2. Our data suggest that the percentage of peak heart rate for VT1 is approximately 70% with a range of 60–80%, while VT2 is approximately 88% with a range of 80–95%. Additionally, our data suggest that the percentage of maximum workload for VT1 is approximately 50% with a range of 40–60%, while VT2 is approximately 75% with a range of 60–90%. Table 7 compares the findings for ranges of this study with ranges from other studies. The differences in previously reported VT1 and VT2 ranges compared to the ranges found in the current study may be explained by the heterogenous fitness of the subjects in the current study; a homogenous fitness population would create smaller ranges for heart rate and workload VT1 and VT2, while a heterogenous population would create larger ranges. Furthermore, previous studies have found that with exercise training, VT1 and VT2 occurs at greater percentages of VO_2_max [28,29,30]. This suggests that individuals with lower fitness, defined by VO_2_max, achieve VT1 and VT2 at lower percentages of their maximal heart rate and workload intensities. However, in recovering stroke patients, heart rate at the ventilatory threshold was predicted and found to be approximately 80% of peak heart rate in a symptom limited GXT, suggesting illness can alter this relationship [6]. As such, these factors can account for the differences in reported ranges for workload and heart rate at VT1 and VT2.

### 4.8. Limitations

In this study, there are several limitations that should be mentioned. This includes requesting but not controlling for subjects’ pre- and post-testing lifestyle behaviors. Not controlling for these extraneous lifestyle behaviors and variables such as diet, sleep, and prior exercise training could have contributed to the increased ranges for VT1 and VT2. Furthermore, we did not control for the time of day that testing took place. Previous studies had found that circadian rhythm could affect maximal aerobic power and time to exhaustion [31,32]. Additionally, we are unaware of the inter-unit variability for measurement error. As such, the validity data could be affected by the different smart shirts used. Finally, the heterogeneity of subjects’ fitness levels may have contributed to the wider ranges for VT1 and VT2 found in the current study compared to the ranges in literature. With this in mind, further studies may look to examine the efficacy of the smart shirt to detect VT1 and VT2 in cohorts that have homogenous fitness.

## 5. Conclusions

The Tyme Wear smart shirt allows the opportunity for users to have relatively affordable access to information that was once only obtainable through a laboratory with the appropriate equipment and typically more expensive cost of testing. However, the caveat is that the detection of VT1 and VT2 by the Tyme Wear smart shirt, at its current stage in development, is less valid with similar reliability to its laboratory counterpart. However, the suggested ranges for exercise intensities for VT1 and VT2 should be an adequate workload for most individuals, although caution should be taken with individuals with high fitness. As such, the Tyme Wear smart shirt provides easily accessible testing to establish threshold-guided training zones but is unlikely to replace or devalue the need for the laboratory equivalent.

## Figures and Tables

**Figure 1 ijerph-19-01147-f001:**
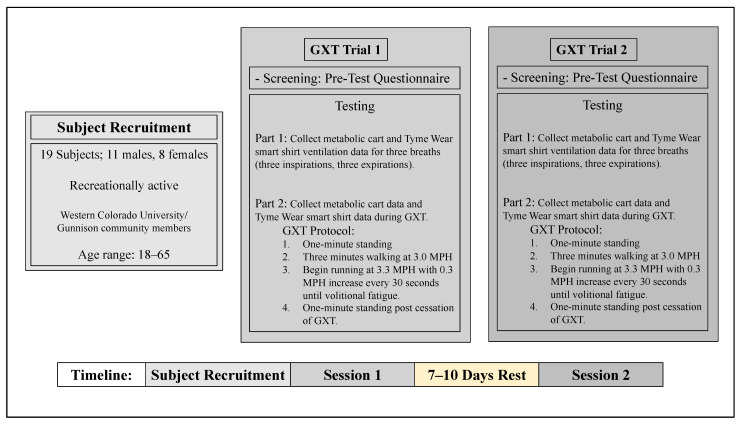
Experimental flow chart outlining study design. GXT, graded exercise test; MPH, miles per hour; WCU, Western Colorado University.

**Figure 2 ijerph-19-01147-f002:**
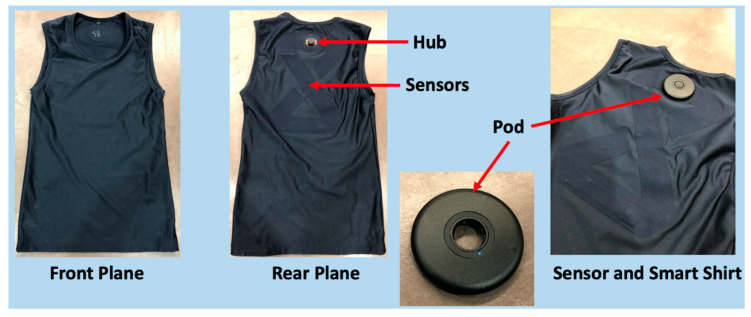
External Layout of Tyme Wear Smart Shirt and Pod.

**Figure 3 ijerph-19-01147-f003:**
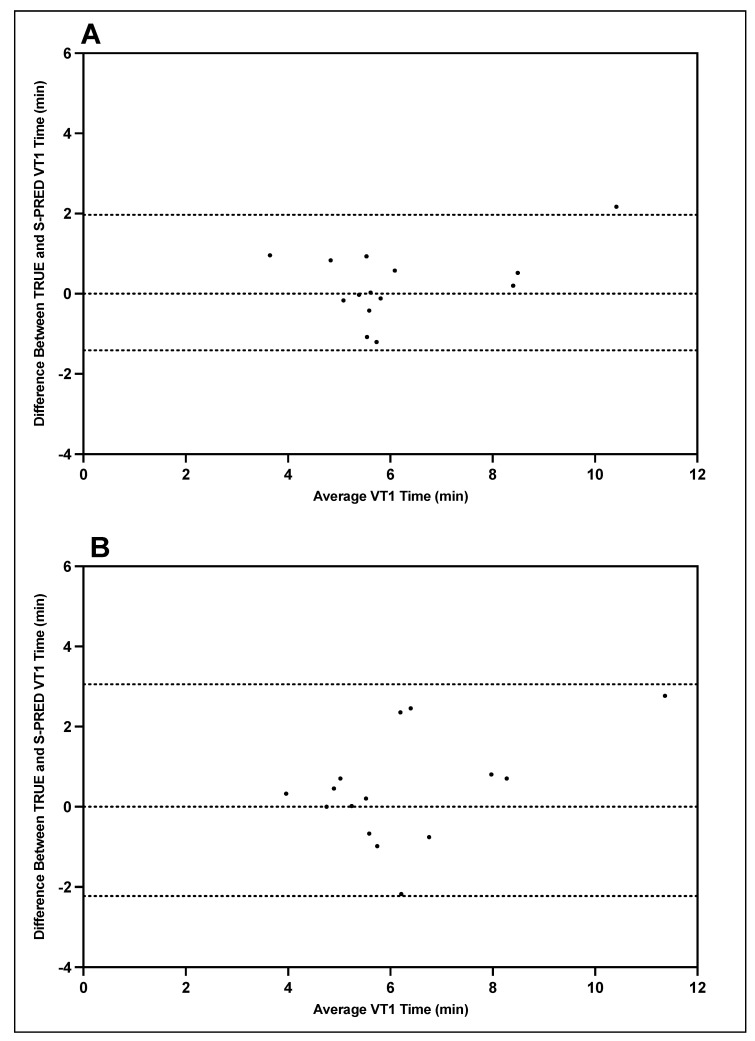
TRUE versus S-PRED VT1 time point; trial 1 (**A**) and trial 2 (**B**).

**Figure 4 ijerph-19-01147-f004:**
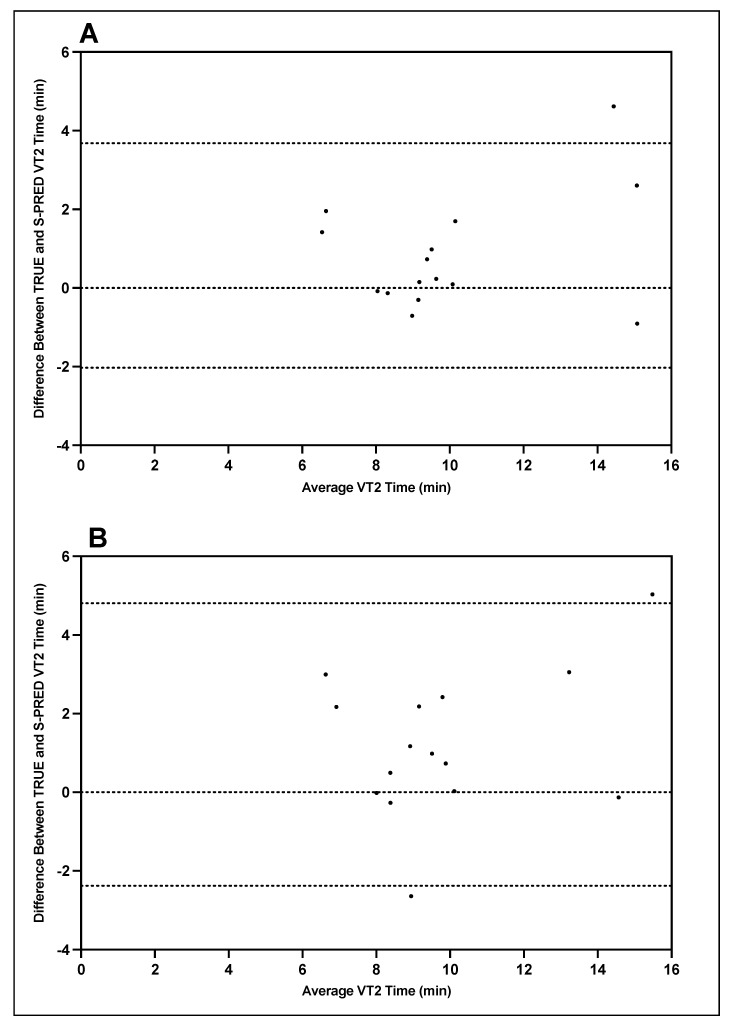
TRUE versus S-PRED VT2 time point; trial 1 (**A**) and trial 2 (**B**).

**Figure 5 ijerph-19-01147-f005:**
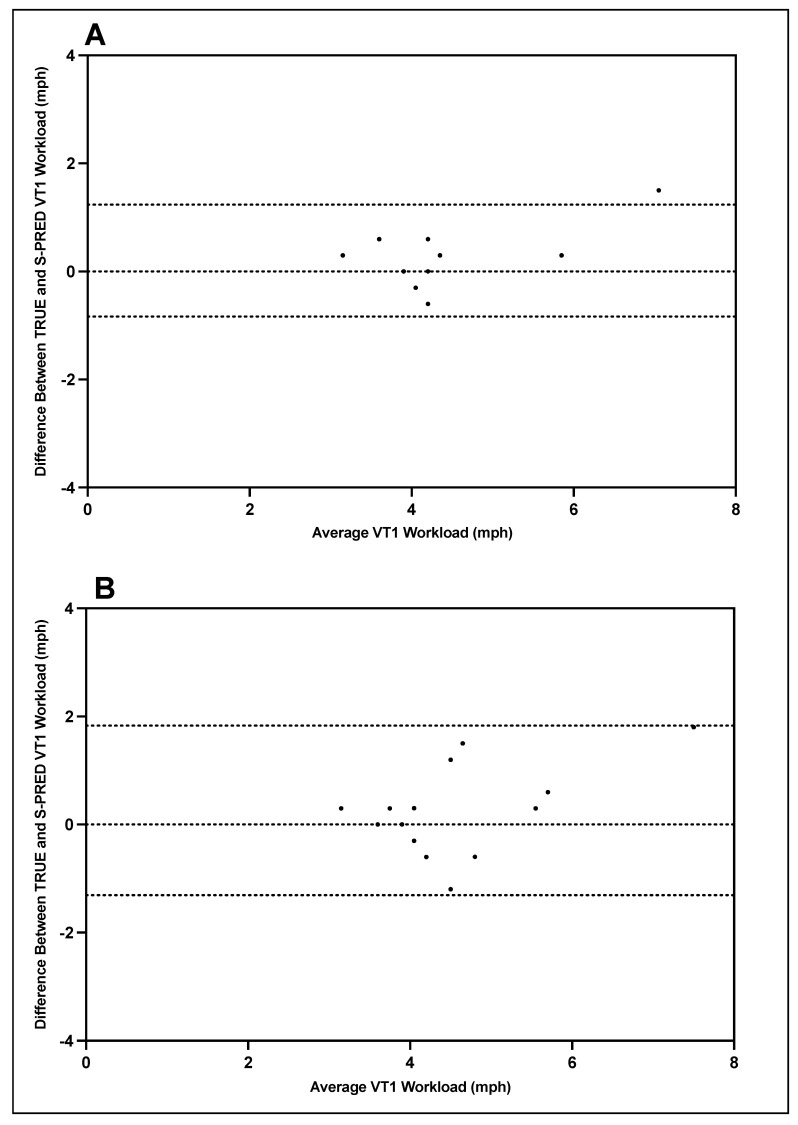
TRUE versus S-PRED VT1 workload; trial 1 (**A**) and trial 2 (**B**).

**Figure 6 ijerph-19-01147-f006:**
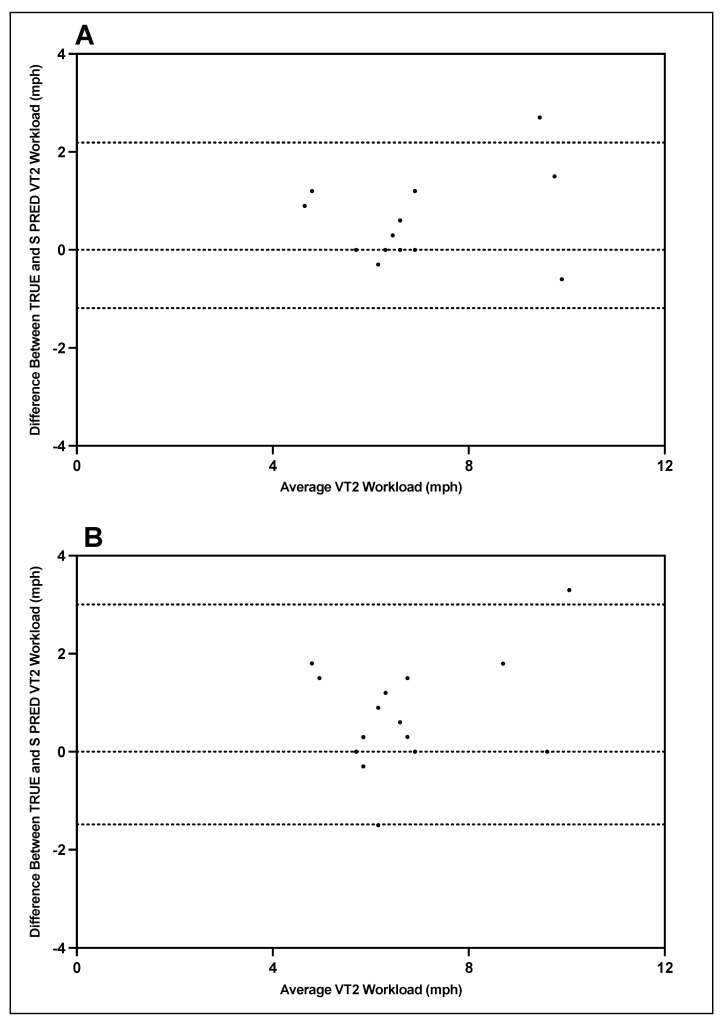
TRUE versus S-PRED VT2 workload; trial 1 (**A**) and trial 2 (**B**).

**Figure 7 ijerph-19-01147-f007:**
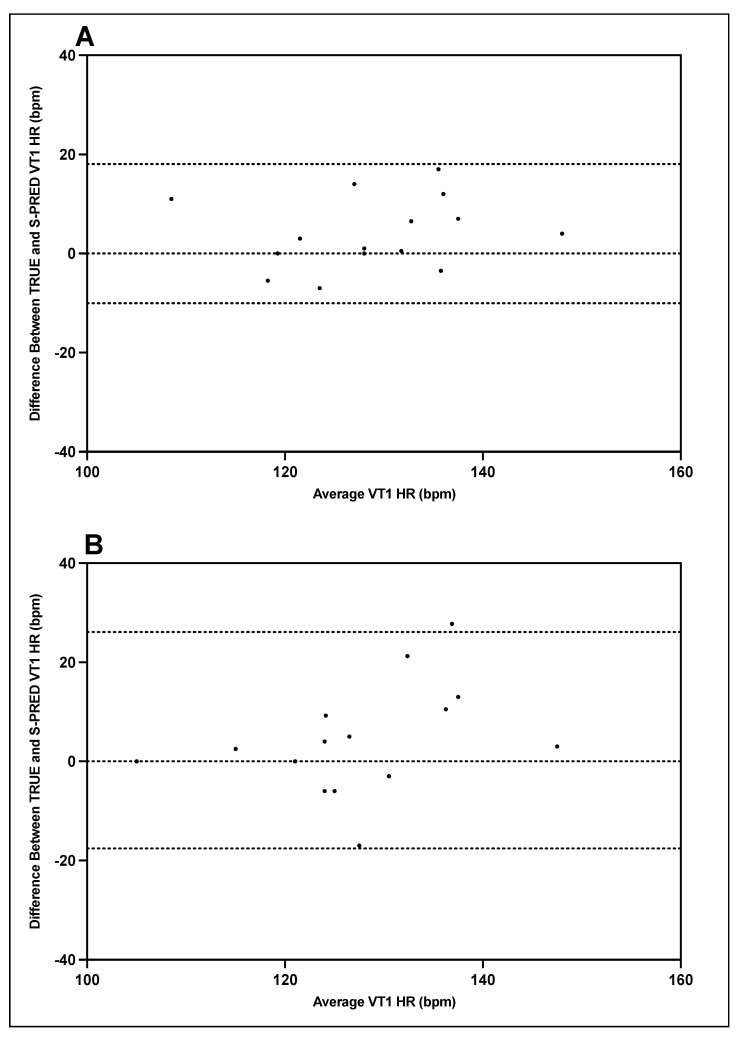
TRUE versus S-PRED VT1 heart rate; trial 1 (**A**) and trial 2 (**B**).

**Figure 8 ijerph-19-01147-f008:**
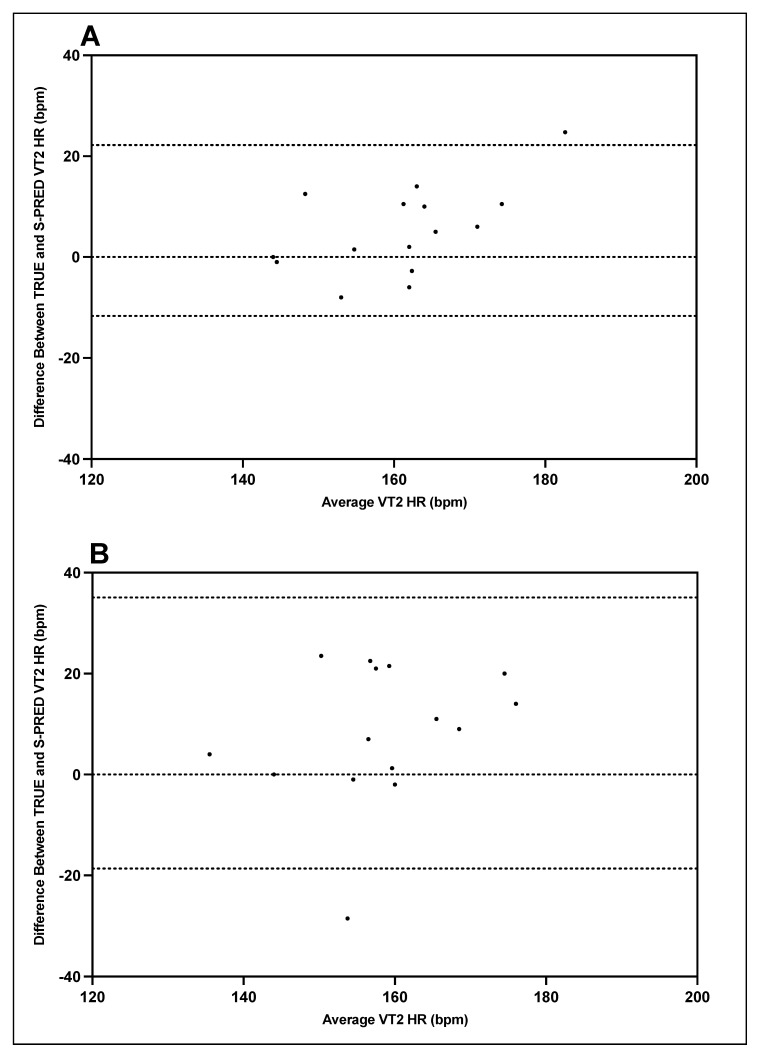
TRUE versus S-PRED VT2 heart rate; trial 1 (**A**) and trial 2 (**B**).

**Table 1 ijerph-19-01147-t001:** Demographics and anthropometrics of subjects.

Subjects	Age (Years)	Height (cm)	Body Mass (kg)	VO_2_max (mL⋅kg^−1^⋅min^−1^)	Peak Heart Rate (bpm)
Total (*n* =15)	27.7 ± 8.4	170.4 ± 8.4	67.8 ± 9.5	48.4 ± 10.5	184.6 ± 8.9
Male (*n* = 8)	29.0 ± 9.5	173.1 ± 8.5	73.1 ± 8.1	52.8 ± 12.4	183.8 ± 11.1
Female (*n* = 7)	26.1 ± 7.5	167.3 ± 7.8	61.6 ± 7.1	43.4 ± 5.1	185.5 ± 5.6

Values are mean ± SD.

**Table 2 ijerph-19-01147-t002:** Measures of reliability for TRUE parameters: CV, TE, and ICC.

Parameters	CV (%)	TE	ICC
TRUE VT1			
Time	8.0	0.25	0.95
HR	3.2	0.35	0.90
Workload	6.8	0.25	0.95
TRUE VT2			
Time	5.3	0.22	0.96
HR	2.4	0.29	0.93
Workload	4.9	0.22	0.96

Units: time (minutes), HR (bpm), workload (mph). CV, coefficient of variation; HR, heart rate; ICC, intraclass correlation coefficient; TE, typical error; VT1, ventilatory threshold 1; VT2, ventilatory threshold 2.

**Table 3 ijerph-19-01147-t003:** Measures of reliability for S-PRED parameters: CV, TE, and ICC.

Parameters	CV (%)	TE	ICC
S-PRED VT1			
Time	9.1	0.34	0.91
HR	3.9	0.58	0.78
Workload	6.9	0.33	0.91
S-PRED VT2			
Time	4.6	0.17	0.98
HR	2.8	0.48	0.84
Workload	4.3	0.18	0.97

Units: time (minutes), HR (bpm), workload (mph). CV, coefficient of variation; HR, heart rate; ICC, intraclass correlation coefficient; TE, typical error; VT1, ventilatory threshold 1; VT2, ventilatory threshold 2.

**Table 4 ijerph-19-01147-t004:** Time point, workload, and heart rate at VT1 and VT2 for TRUE versus S-PRED during trial 1 and trial 2.

Parameter	Trial 1		Trial 2	
	TRUE	S-PRED	TRUE	S-PRED
VT1				
Time point (min)	6.3 ± 1.9 ^a^	6.0 ± 1.6	6.5 ± 2.2	6.1 ± 1.6
Workload (mph)	4.6 ± 1.2	4.4 ± 0.9	4.6 ± 1.3	4.4 ± 1.0
Heart rate (bpm)	130.8 ± 10.9	126.8 ± 9.7 *	129.7 ± 13.3	125.4 ± 9.6
VT2				
Time point (min)	10.5 ± 3.0	9.1 ± 2.6 *	10.5 ± 3.0	9.3 ± 2.5 *
Workload (mph)	7.1 ± 1.8	6.6 ± 1.6 *	7.1 ± 1.8	6.4 ± 1.5 *
Heart rate (bpm)	163.5 ± 13.6	158.2 ± 8.9 *	162.3 ± 14.1	154.0 ± 10.89 *

^a^ Values are mean ± SD; * denotes significant between-group difference. VT1, ventilatory threshold 1; VT2, ventilatory threshold 2.

**Table 5 ijerph-19-01147-t005:** VT1 and VT2 as a percentage of peak heart rate.

Parameter	Trial 1		Trial 2	
	TRUE	S-PRED	TRUE	S-PRED
VT1				
Average % of Max	71 ± 5 ^a^	69 ± 5 *	70 ± 7	68 ± 5
Maximum HR %	81	78	85	77
Minimum HR %	63	58	57	57
VT2				
Average % of Max	88 ± 5	86 ± 3 *	88 ± 7	83 ± 5 *
Maximum HR %	96	90	99	88
Minimum HR %	78	81	72	76

^a^ Values are mean ± SD; * denotes significant between-group difference. VT1, ventilatory threshold 1; VT2, ventilatory threshold 2.

**Table 6 ijerph-19-01147-t006:** VT1 and VT2 as a percentage of achieved maximum workload speed (mph).

Parameter	Trial 1		Trial 2	
	TRUE	S-PRED	TRUE	S-PRED
VT1				
Average % of Max	48 ± 5 ^a^	46 ± 2	49 ± 8	46 ± 4
Maximum mph %	60	50	64	53
Minimum mph %	40	40	38	38
VT2				
Average % of Max	74 ± 7	69 ± 4 *	75 ± 9	66 ± 5 *
Maximum mph %	88	79	89	76
Minimum mph %	63	60	56	59

^a^ Values are mean ± SD; * denotes significant between-group difference. VT1, ventilatory threshold 1; VT2, ventilatory threshold 2.

**Table 7 ijerph-19-01147-t007:** Comparison of VT1 and VT2 workload and heart rate ranges.

	Data from TRUE	Cerezuela-Espejo et al.	Boyne et al.	Gasparini Neto et al.
Population Type	Recreationally Active	Trained Athletes	Chronic Stroke Patients	Non-Trained| Elite Athletes
VO_2_max or peak (mL·kg^−1^·min^−1^)	48.4 ± 10.5	60.2 ± 4.3	16.2 ± 5.7	47.2 ± 4.4| 68.6 ± 3.2
Workload VT1 (% of Max Speed)	48.5 ± 6.5 (Range: 39–62)	59–65	-	57.1 ± 6.9| 70.0 ± 6.8
Workload VT2 (% of Max Speed)	74.5 ± 8.0 (Range: 59.5–88.5)	84–87	-	80.4 ± 9.0| 87.7 ± 4.1
Heart Rate VT1 (% of Max or Peak Heart Rate)	70.5 ± 6.0 (Range: 60–83)	77–81	80.0 ± 8.0 (Range: 68–94)	76 ± 5.4| 85.5 ± 4.2
Heart Rate VT2 (% of Max or Peak Heart Rate)	88 ± 6.0 (Range: 75–97.5)	91–93	-	91.4 ± 4.0| 94.9 ± 2.4

Values are mean ± SD.

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
