# Peer review of "Is the Tyme Wear Smart Shirt Reliable and Valid at Detecting Personalized Ventilatory Thresholds in Recreationally Active Individuals?"

_ijerph, 2022, doi:10.3390/ijerph19031147_

Round 1
Reviewer 1 Report
The main questions addressed by the research is the validity of smart shirt for evaluation CPET, that means cardiorespiratore exrecise testing and the comparison with traditional testing by classical method for gas exchange measuring. The result are rathter good thus promoting smart shirt for exerise testing in a larger papulation. The topic is very relevant as the method as such can be used by a broader number of physicians without expensive technique. This is a new method using wearbale technology and may be wide spread in the future, and, further application, can be used in field test under sports related situations.The authors should give a short comment if there are gender differences due to anatomical reasons.
Das age glay a role for validity ? There are some statistical outliers presented in the figures, the authors should give a short comment on this and explain.
The authors should test the smart shirt in a larger population with males and females and even in elderly athletes.
Author Response
Reviewer #1
The main questions addressed by the research is the validity of smart shirt for evaluation CPET, that means cardiorespiratory exercise testing and the comparison with traditional testing by classical method for gas exchange measuring. The result are rather good thus promoting smart shirt for exercise testing in a larger papulation. The topic is very relevant as the method as such can be used by a broader number of physicians without expensive technique. This is a new method using wearable technology and may be wide spread in the future, and, further application, can be used in field test under sports related situations.
We appreciate the kind remarks and careful review from Reviewer #1. We also agree with the suggested future line of research, which we are already pursuing in our laboratory.
The authors should give a short comment if there are gender differences due to anatomical reasons.
There were no gender differences which is why the reliability and validity analyses are pooled.
Does age play a role for validity? There are some statistical outliers presented in the figures, the authors should give a short comment on this and explain.
No, age did not play a role in validity. Although to truly address this question a separate cohort with a larger age range would be needed.
However, as you highlight with your comment, there were some subjects where the difference between S-PRED and TRUE were more pronounced. We have provided a comment and potential explanation for this on page 17 (lines 372-380):
“Upon closer inspection of the Bland Altman plots, there are four particular subjects whose underestimation are consistently larger than the others for workload and heartrate at VT1 and VT2 across both trials 1 and 2. Specifically, in Figure 6 and Figure7, it appears that these four subjects that had the greatest difference between S-PRED and TRUE also had higher fitness levels (VO2max > 60 mL⋅kg-1⋅min-1 for males and > 50 mL⋅kg-1⋅min-1 for females) when compared to the rest of the cohort, whereby VT1 and VT2 workload occur at higher intensities. One possible explanation for this is that the algorithms used to model VT1 and VT2 are not well suited for those users with greater fitness levels and may be unable to accurately detect VT1 and VT2 workloads for these types of individual.”
The authors should test the smart shirt in a larger population with males and females and even in elderly athletes.
Thank you for this suggestion. We agree and are planning these studies for the near future.
Reviewer 2 Report
Dear Editor, Thank you for the opportunity to review this manuscript aimed to determine the reliability and validity of the TymWear smart shirt compared to the Parvo Medics TrueOne 2400 in determining ventilatory thresholds in recreationally active men and women.
TITLE
The title of the manuscript must be modified. A possible title of the present article should be… Is the Tyme Wear Smart Shirt Reliable and Valid at Detecting Personalized Ventilatory Thresholds in Recreationally Active Individuals?
INTRODUCTION
The introduction argues well the topics covered while included all the relevant references.
MATERIALS AND METHODS
The methods applied adequately describe the authors' work, as well as the inclusion and exclusion criteria, have been precisely chosen.
RESULTS AND DISCUSSION
The results are written clearly and readable way for the reader.
The discussion correctly argues the results identified.
STRENGTHS and LIMITATIONS
In the present study, the authors focused to evaluate the reliability and validity of the Tyme Wear smart shirt to predict VT1 and VT2 when compared to the gold standard determined by open-circuit indirect calorimetry, however, no strengths were highlighted. In order to improve the quality and the visibility of the present study please add a paragraph regarding the strengths and of this research study.
CONCLUSIONS
The conclusions written are supported by the results obtained.
I see the potential contribution of this manuscript which is of scientific interest and is in line with the aims of the journal. The author's guidelines have been respected and the manuscript does not require a revision of the English language and may be accepted for publication in the International Journal of Environmental Research and Public Health.
Author Response
Reviewer #2
Dear Editor, Thank you for the opportunity to review this manuscript aimed to determine the reliability and validity of the TymWear smart shirt compared to the Parvo Medics TrueOne 2400 in determining ventilatory thresholds in recreationally active men and women.
We appreciate the kind remarks and careful review from Reviewer #2. We have made all the suggested revisions from Reviewer #2. Thank you.
TITLE
The title of the manuscript must be modified. A possible title of the present article should be… Is the Tyme Wear Smart Shirt Reliable and Valid at Detecting Personalized Ventilatory Thresholds in Recreationally Active Individuals?
We have revised the title as you have suggested.
INTRODUCTION
The introduction argues well the topics covered while included all the relevant references.
Thank you.
MATERIALS AND METHODS
The methods applied adequately describe the authors' work, as well as the inclusion and exclusion criteria, have been precisely chosen.
Thank you.
RESULTS AND DISCUSSION
The results are written clearly and readable way for the reader.
The discussion correctly argues the results identified.
Thank you.
STRENGTHS and LIMITATIONS
In the present study, the authors focused to evaluate the reliability and validity of the Tyme Wear smart shirt to predict VT1 and VT2 when compared to the gold standard determined by open-circuit indirect calorimetry, however, no strengths were highlighted. In order to improve the quality and the visibility of the present study please add a paragraph regarding the strengths and of this research study.
Per your suggestion we have revised the first paragraph of section 4.7 on page 18 to better highlight the strengths of the study:
4.7. Strength of Study and Additional Findings
The primary strength of the present study is it is the first investigation, to our knowledge, of the reliability and validity of a wearable technology that measures respiratory rate and tidal volume to calculate VE in order to detect VT1 and VT2. Our novel findings provide the necessary preliminary data to establish the potential for the Tyme Wear smart shirt to improve performance by optimizing training with the use of personalized VT1 and VT2.
CONCLUSIONS
The conclusions written are supported by the results obtained.
I see the potential contribution of this manuscript which is of scientific interest and is in line with the aims of the journal. The author's guidelines have been respected and the manuscript does not require a revision of the English language and may be accepted for publication in the International Journal of Environmental Research and Public Health
Thank you – we appreciate your review of our paper.
Reviewer 3 Report
In the paper, the author has made a good attempt to evaluate the reliability and validity of the Tyme Wear Shirt. The abstract of the paper is well written, covering the purpose and findings of the study. However, in my opinion, the end part can be written more precisely like the statement at line number 22 “The result from this study suggests that the Tyme Wear smart shirt is less valid but is comparable in reliability when compared to 23 the gold standard” is vague. The method and Materials section includes a fair description. However, the result section can be improved by adding a few more useful graphs.
- At lines 16 to 21 a summary of the results is mentioned using acronyms (ICC, p-value) that are not described. Also, in line 244, “with 95% CIs are” but Cis is not defined earlier.
- The author may elaborate on the ventilatory threshold 1, 2 (section 1. Introduction) and the functional relationship with “volume of oxygen consumed (VO2) or VE” the parameter which is measured by the device primarily.
- The author can elaborate on the pre-test questionnaire (in section 2.2 Procedure). The author can include the questionnaire in the text or an appendix. Also, does the study include a questionnaire or exit interview on the ergonomics of the wearable shirt?
- The author can elaborate on the exclusion criteria of the participants.
- Few more pictures or figures can be included to demonstrate the experimental scenario and setup.
- The author has used Microsoft Excel software for data analysis. However, the author may use more effective and efficient tools like MATLAB for determining statistical parameters. The author has included the reference for statistical analysis. However, it would be good if the author includes the formula and explanation here as well. It would be good to elaborate on paired t-tests, or references can be included.
- The results can be displayed more effectively. The author has plotted the difference between the average value for individual measurements. To better justify the validity of measurements, the Box plot or five-point summary plot can be shown for all the data of each participant to better summarize the results.
- The data presented in plots need to be tagged with the participant to find out the boundary conditions for confidence intervals is referenced to any particular participant in all trials, or it is random.
- Line 221 need to be re-written either mentioning the null hypothesis or describing the details of what it means to have no significant comparisons? (“Other TRUE versus S-PRED parameter comparisons were not significant (p > 0.05)”)
- In line 307, the author needs to specifically mention the percentage improvement achieved with reference to the earlier work. (“The finding from the current study for 306 CV VT1 workload is also similar to that suggested by Dickhuth and colleagues (CV = 307 5.8%) [19]”)
- Authors need to improve on grammatical mistakes and typo errors. For instance, at line no 151.
Author Response
Reviewer #3
In the paper, the author has made a good attempt to evaluate the reliability and validity of the Tyme Wear Shirt.
We appreciate the kind remarks and careful review from Reviewer #3. We have revised our paper accordingly. In most instances, we agree with the suggestions provided by Reviewer #3. Where we have elected to not take up the suggested revision, we have provided our rationale for not making the edit.
The abstract of the paper is well written, covering the purpose and findings of the study. However, in my opinion, the end part can be written more precisely like the statement at line number 22 “The result from this study suggests that the Tyme Wear smart shirt is less valid but is comparable in reliability when compared to 23 the gold standard” is vague.
We have reviewed this section of the abstract and feel the current wording accurately captures our findings. Since neither Reviewer #1 nor Reviewer #2 had any comments on the abstract wording, we have elected to leave unchanged.
The method and Materials section includes a fair description. However, the result section can be improved by adding a few more useful graphs.
As mentioned previously we have endeavored to revise our paper in most instances as you have suggested.
1. At lines 16 to 21 a summary of the results is mentioned using acronyms (ICC, p-value) that are not described. Also, in line 244, “with 95% CIs are” but Cis is not defined earlier.
We have spelled out ICC and CI before abbreviating as you have suggested. Thank you.
2. The author may elaborate on the ventilatory threshold 1, 2 (section 1. Introduction) and the functional relationship with “volume of oxygen consumed (VO2) or VE” the parameter which is measured by the device primarily.
Although we appreciate your comment, we have elected not to elaborate here (as you have suggested) as this topic is highlighted in one of our previous papers:
Gouw, A.H.; Van Guilder, G.P.; Larusson, A.; Laredo, G.; Weatherwax, R.M.; Byrd, B.R.; Dalleck, L.C. Ventilation can exclusively be used to predict ventilatory thresholds: a retrospective analysis. Int. J. Res. Ex. Phys. 2021, 16, 1–18.
3. The author can elaborate on the pre-test questionnaire (in section 2.2 Procedure). The author can include the questionnaire in the text or an appendix. Also, does the study include a questionnaire or exit interview on the ergonomics of the wearable shirt?
We have added the pre-test questionnaire as Appendix A as you have suggested.
We did not have an exit interview question on the ergonomics of the smart shirt. This is an excellent suggestion and we will incorporate into future research.
4. The author can elaborate on the exclusion criteria of the participants.
There is not much to elaborate on beyond what is currently listed. Individuals with cardiovascular and/or pulmonary conditions/diseases were excluded as they generally require medical supervision for maximal exercise testing. We also excluded individuals who were unable to run on the treadmill to volitional fatigue (as indicated in the wording of our originally submitted paper).
5. Few more pictures or figures can be included to demonstrate the experimental scenario and setup.
No additional images or figures were added. We did not feel this would add substantial to the paper. Readers can readily look online to see what the smart shirt looks like on a user. Additionally, the VO2max set up is extremely well known in the exercise science community.
6. The author has used Microsoft Excel software for data analysis. However, the author may use more effective and efficient tools like MATLAB for determining statistical parameters. The author has included the reference for statistical analysis. However, it would be good if the author includes the formula and explanation here as well. It would be good to elaborate on paired t-tests, or references can be included.
Thank you for this comment. We used a combination of Microsoft Excel and Graphpad Prizm statistical software for all analyses. Our team does not use MATLAB and we feel its use is more of a personal preference for data analyses rather than a tool/strategy which might change the findings. As such, we have left this section unchanged.
7. The results can be displayed more effectively. The author has plotted the difference between the average value for individual measurements. To better justify the validity of measurements, the Box plot or five-point summary plot can be shown for all the data of each participant to better summarize the results.
Similar to the previous comment, we appreciate the thoughts of the reviewer here. However, we feel it’s more an issue of personal preference (of the reviewer) on how to present the Bland Altman plots versus one plot being superior to the other. We have followed standardized guidelines (Bland and Altman, 1986) for our plots and also followed formatting we have done previously:
Gouw, A.H.; Van Guilder, G.P.; Larusson, A.; Laredo, G.; Weatherwax, R.M.; Byrd, B.R.; Dalleck, L.C. Ventilation can exclusively be used to predict ventilatory thresholds: a retrospective analysis. Int. J. Res. Ex. Phys. 2021, 16, 1–18.
8. The data presented in plots need to be tagged with the participant to find out the boundary conditions for confidence intervals is referenced to any particular participant in all trials, or it is random.
We do not feel this is necessary as our current paper permits identification of those individuals where there was a more pronounced difference between S-PRED and TRUE, and we comment on that on page 17 (lines 372-380):
“Upon closer inspection of the Bland Altman plots, there are four particular subjects whose underestimation are consistently larger than the others for workload and heartrate at VT1 and VT2 across both trials 1 and 2. Specifically, in Figure 6 and Figure7, it appears that these four subjects that had the greatest difference between S-PRED and TRUE also had higher fitness levels (VO2max > 60 mL⋅kg-1⋅min-1 for males and > 50 mL⋅kg-1⋅min-1 for females) when compared to the rest of the cohort, whereby VT1 and VT2 workload occur at higher intensities. One possible explanation for this is that the algorithms used to model VT1 and VT2 are not well suited for those users with greater fitness levels and may be unable to accurately detect VT1 and VT2 workloads for these types of individual.”
Additionally, from our experience, it would be unusual and cluttered to provide individual labels for all the participants on each plot. In summary, we feel this revision is unnecessary and would, if anything, make the figures less impactful in presenting our key findings.
9. Line 221 need to be re-written either mentioning the null hypothesis or describing the details of what it means to have no significant comparisons? (“Other TRUE versus S-PRED parameter comparisons were not significant (p > 0.05)”)
Thank you for your point. After careful review, we have stated what the significant differences are in lines 218-225. Therefore, we have elected to omit this sentence for simplicity.
10. In line 307, the author needs to specifically mention the percentage improvement achieved with reference to the earlier work. (“The finding from the current study for 306 CV VT1 workload is also similar to that suggested by Dickhuth and colleagues (CV = 307 5.8%) [19]”)
Our aim here was not to demonstrate a percentage improvement but rather to compare/contrast our findings to those previously reported in the scientific literature. The CV we achieved were consistent with what has been previously done and are well below 10%. Please advise if we have misunderstood you.
11. Authors need to improve on grammatical mistakes and typo errors. For instance, at line no 151.
Corrected the error here as you highlighted. Thank you. We have reviewed the remainder of the paper a few times and haven’t found any additional mistakes.
This manuscript is a resubmission of an earlier submission. The following is a list of the peer review reports and author responses from that submission.
Round 1
Reviewer 1 Report
Wearable sensors are becoming popular and need to be physiologically validated by independent scientists. Ventilatory threshold detection is an appropriate metric to assess correspondence or not between a metabolic cart and the Tyme Wear Smart Shirt. Thus, there is imperative for this study which, to a point, is well designed. However, there are serious concerns that undermine enthusiasm for this work. Paramount among these concerns are:
- The V-slope was developed at Harbor-UCLA by Beaver, Whipp and Wasserman (1986) to overcome the problems inherent with ventilation threshold detection using just VE or ventilatory equivalents. A strong case can be made for the “gold standard” for such being GET measured by V-slope (and even validated by measuring blood lactate).
- This is written in very biased, judgmental and subjective language. Develop testable hypotheses (as alluded to in Methods) in the Introduction and let the data speak for themselves.
- From #2 where are the actual data. The BA plots are boring and do not help convince the reader that GET, VT1 or VT2 can be reliably measured/discerned in these investigators hands.
- GET really only becomes a parameter when measured as VO2. This is because, depending on rate of WR increase/speed ramp the VO2 at which GET occurs remains unchanged even if the WR/speed changes – which it does (Davis et al. 1982). Thus report data accordingly. Also, error as VO2 would be helpful.
- Throughout the sections are far longer than necessary: reduce by at least 1/3.
- Methods: Although VO2max is not a primary outcome, without a validation test, these criteria (with the exception of the demonstrated plateau) do not provide evidence for VO2max in many subjects.
- As Critical Power/Critical Speed has arguably greater relevance to sports performance than VT1 or VT2 would this be a better metric to measure – or at least relate your findings to (Jones et al. 2019)? The error term then could be incorporated into further evaluation as exercise performed above vs below CP/CS has a markedly different metabolic and performance outcome. The 13% mph difference between TRUE and S-PRED is thus extremely concerning.
- It is in here but please use test/retest reliability terminology (with mean delta or CV presented).
- Discussion first para: similar (no! and not tested), less valid, encouraging. Next para: “excellent” – all are overly judgemental and subjective.
- Line 301 How can you separate technical error from biological variability?
- Line 341 Not physiologically significant! Yes, it can be.
- Lines 341-5 Depends what YOU set as the “range.”
- Line 373 Need more information on those algorithms and, perhaps, some actual data as regards VE on the test. At some point VE or delta relative VE or something needs to be directly compared between systems. This needs considering beyond the explanation given in lines 383-
- Line 404 needs referencing explicitly. Paragraph 4.6 might mention that CP/CS require only a stop watch for measurement.
- Line 448 Aging has this compressive effect as well as certain diseases.
- Line 471 By reliability is reproducibility meant? They are very different concepts.
Beaver WL, Wasserman K, Whipp BJ. A new method for detecting anaerobic threshold by gas exchange. J Appl Physiol (1985). 1986 Jun;60(6):2020-7. doi: 10.1152/jappl.1986.60.6.2020. PMID: 3087938.
Davis JA, Whipp BJ, Lamarra N, Huntsman DJ, Frank MH, Wasserman K. Effect of ramp slope on determination of aerobic parameters from the ramp exercise test. Med Sci Sports Exerc. 1982;14(5):339-43. PMID: 7154888.
Jones AM, Burnley M, Black MI, Poole DC, Vanhatalo A. The maximal metabolic steady state: redefining the 'gold standard'. Physiou Rep. 2019 May; 7(10):e14098. doi 10.14814/phy2.14098. PMID:31124324; PMCID: PMC6533178
Poole DC, Wilkerson DP, Jones AM. Validity of criteria for establishing maximal O2 uptake during ramp exercise tests. Eur J Appl Physiol. 2008 Mar;102(4):403-10. doi: 10.1007/s00241-007-0596-3. Epub 2007 Oct 30. PMID: 17968581.
Reviewer 2 Report
The aim of this study was to evaluate the accuracy and reliability of the Tyme Wear smart shirt in detecting ventilatory threshold. The study design was appropriate and well-conducted. My primary concern for this study are related to the statistical assessment and representation of the results. Details are shown below.
Abstract: if there's room, please show some data in the abstract
Stats: a paired samples t-test is a test of differences, and does not show if values are similar. Rather, in assessing accuracy of a wearable, it is best to conduct an equivalence test. Please substitute the t-test for an equivalence test
Results: I love the inclusion of the Bland-Altman plots, but I am not sure that separate plots for each trial need to be included. Perhaps using colored figures would allow placing data from GXT1 and GXT2 on the same figures and the data could be combined to calculated new limits of agreement.